# The Use of Cardiac Autonomic Responses to Aerobic Exercise in Elderly Stroke Patients: Functional Rehabilitation as a Public Health Policy

**DOI:** 10.3390/ijerph182111460

**Published:** 2021-10-31

**Authors:** Rodrigo Daminello Raimundo, Juliana Zangirolami-Raimundo, Claudio Leone, Tatiana Dias de Carvalho, Talita Dias da Silva, Italla Maria Pinheiro Bezerra, Alvaro Dantas de Almeida, Vitor Engracia Valenti, Luiz Carlos de Abreu

**Affiliations:** 1Faculdade de Saúde Pública, Universidade de São Paulo, Av. Dr. Arnaldo, 715-Cerqueira César, Sao Paulo 01246-000, Brazil; rodrigo.raimundo@fmabc.br (R.D.R.); leone.claudio@gmail.com (C.L.); 2Laboratório de Delineamento de Estudos e Escrita Científica, Centro Universitário FMABC, Av. Lauro Gomes, 2000-Vila Sacadura Cabral, Santo Andre 09060-870, Brazil; juliana.zangirolami@fmabc.br; 3Disciplina de Ginecologia, Departamento de Obstetrícia e Ginecologia, Hospital das Clínicas, Faculdade de Medicina FMUSP, Universidade de São Paulo, Av. Dr. Arnaldo, 455-Cerqueira César, Pacaembu-SP 01246-903, Brazil; 4Departamento de Ciencias de la Salud, Universidad Nacional de La Matanza, Florencio Varela 1903, San Justo B1754, Argentina; carvalho.td1@gmail.com; 5Departamento de Cardiologia, Universidade Federal de São Paulo, Rua Sena Madureira, 1500-1º Andar-Vila Clementino, Sao Paulo 04021-001, Brazil; ft.talitadias@gmail.com; 6Escola Superior de Ciências da Santa Casa de Misericórdia de Vitória, Avenida Nossa Senhora da Penha, 2190-Bela Vista, Vitoria 29027-502, Brazil; italla.bezerra@emescam.br; 7Programa de Pós-Graduação em Ciencias Médicas, Faculdade de Medicina da USP, Universidade de São Paulo, Av. Dr. Arnaldo, 455-Cerqueira César, Pacaembu-SP 01246-903, Brazil; alvaro.dantas@usp.br; 8Centro de Estudos do Sistema Nervoso Autônomo (CESNA), Universidade Estadual Paulista, Av. Hygino Muzzi Filho, 737, Marilia 17525-900, Brazil; vitor.valenti@gmail.com; 9School of Medicine, University of Limerick, Castletroy, V94 T9PX Limerick, Ireland; 10Departamento de Educação Integrada em Saúde, Universidade Federal do Espírito Santo (UFES), Av. Fernando Ferrari, 514-Goiabeiras, Vitória 29075-910, Brazil

**Keywords:** autonomic nervous system, aging, heart rate variability, exercise, stroke

## Abstract

Background and purpose: The development of public policies must be guided by full knowledge of the health–disease process of the population. Aerobic exercises are recommended for rehabilitation in stroke patients, and have been shown to improve heart rate variability (HRV). Our aim was to compare the cardiac autonomic modulation of elderly stroke patients with that of healthy elderly people during and after an acute bout of aerobic exercise. Methods: A total of 60 elderly people participated in the study (30 in the control group, mean age of 67 ± 4 years; 30 in the stroke group, mean age of 69 ± 3 years). HRV was analyzed in rest—10 min of rest in supine position; exercise—the 30 min of peak exercise; and recovery—30 min in supine position post-exercise. Results: Taking rest and exercises together, for SDNN, RMSSD, pNN50, RRTri, and TINN, there was no difference between the stroke and control groups (*p* = 0.062; *p* = 0.601; *p* = 0.166; *p* = 0.224, and *p* = 0.059, respectively). The HF (ms^2^) was higher and the LF/HF ratio was lower for the stroke group than the control group (*p* < 0.001 and *p* = 0.007, respectively). The SD2 was lower for the stroke group than for the control group (*p* = 0.041). Conclusion: Stroke patients present reduced variability at rest, sympathetic predominance during exercise, and do not return to baseline after the 30 min of recovery, with similar responses found in the healthy elderly group.

## 1. Introduction

It is estimated that in 2050 there will be two billion elderly people in the world—~20% of the world population—and that people aged more than 60 years will exceed the population of young people aged under 15 years [1]. Meanwhile, 17 million people die of heart attacks and strokes every year [2].

The development of public policies must be guided by full knowledge of the health–disease process of the population—especially the elderly. The development of care guidelines for the rehabilitation of elderly stroke patients should be carried out on the basis of discussions with a multidisciplinary group; thus, the use of exercises can be a social policy tool for the planning and development of public health policies. The aging process induces the elderly to be more inclined to the development of cardiovascular, metabolic, and cerebrovascular diseases such as stroke [3]. These changes are linked to an increased susceptibility to sudden death [4].

Additionally, recurrent stroke raises the probability of disability and institutionalization [5,6], with approximately half of the survivors remaining disabled, and it is known that physical inactivity is also one of the risk factors for cardiovascular diseases [2].

Both cardiac diseases [7] and stroke [8,9,10,11,12] are associated with aging and impaired autonomic function. Exercise training and physical activity are recommended for health promotion and rehabilitation in older adults [11] and stroke patients.

Alterations in autonomic regulation may be measured via heart rate variability (HRV) [13]. HRV describes the oscillations in the interval between consecutive heart beats (RR interval). This is a measure that can be used to assess the ANS modulation under physiological and pathological conditions [14]. Changes in HRV patterns provide an indicator of health implications. Higher HRV is a signal of good adaptation of the ANS [15]. There are associations between HRV, baroreceptor sensitivity, stroke severity, early and late complications, dependency, and mortality [16,17,18]. Several studies have shown [2] that a decreased HRV can be associated with increased incidence of cardiovascular diseases due to reduced sympathetic activity and decreased parasympathetic modulation. These changes in autonomic regulation are also related to aging [19,20].

Aerobic exercises have been shown to improve HRV. Hamer et al. [21] showed through HRV that exercise training may also play a role in mediating the inhibition of inflammatory mechanisms, claiming the increased vagal heart rate control to be associated with decreased inflammatory responses [11,21]. Galetta et al. showed that regular physical activity reduces the impairment of sympathetic balance with age, with an increased parasympathetic tone [22], while Raimundo et al. showed a reduced HRV during and at least 30 min after exercise, due to an autonomic imbalance reflected by increased indices that represent the sympathetic nervous system in stroke patients [12].

Eating a healthy, balanced diet accompanied by regular exercise is considered an important intervention for stroke; this association improves insulin sensitivity, blood pressure, and HDL cholesterol levels. Furthermore, muscle weakness represents one of the biggest contributors to disability after stroke [23]. It is important that, at any level of healthcare, exercises are designed so that muscle activity results in greater functional capacity. Guidelines for the rehabilitation of stroke patients are indicated according to the World Report on Disability, published by the World Health Organization (WHO) in 2011 [24]. States must provide effective and appropriate measures to enable people with disabilities to achieve and maintain maximum autonomy and full physical, mental, social, and professional capacity, as well as full inclusion and participation in all aspects of life [25].

HRV is measured by the variation in the beat-to-beat interval, and may contain indicators of current disease or disease predictors [13,15]. Previous studies have shown that aerobic exercises induce a resting bradycardia accompanied by increased cardiac vagal modulation; hence, exercise training may be able to exert an antiarrhythmic effect. In stroke rehabilitation, aerobic exercise provides beneficial effects on cardiovascular capacity as well as emotional condition [6,16,23].

Stroke is the most disabling health problem today [26], according to the World Health Organization (2020); thus, as it is a public health problem that requires the action of a multidisciplinary team in order to provide prevention and promotion of individuals, this highlights the importance of recognizing associated factors that imply the early identification of cardiovascular changes—or, in patients already affected, the identification of their peculiarities that can aggravate the situation—and, finally, designing a better plan for decision making by healthcare professionals [25].

In this context, the understanding of changes in heart rate variability patterns becomes an instrument that will contribute as a clinical modality in the identification of cardiovascular health problems and that, concomitantly with prevention and health-promotion actions, can help to reduce lethality and mortality resulting from this injury [27]. Thus, in the context of public health, studies on the analysis of HRV patterns, based on their analysis models and clinical application, can be an especially valuable tool in the diagnosis and treatment of complications in cardiovascular health, resulting in the reduction in indicators of poor health, as well as becoming a tool for daily use in clinical practice through low-cost and easy-to-apply equipment [24,28].

Thus, the need for effective stroke rehabilitation is likely to remain an essential part of the stroke care continuum for the foreseeable future, in the aim of improving health policies for functional rehabilitation [23]. However, there are few studies comparing an elderly stroke population to healthy elderly control subjects. Our hypothesis is that an acute session of aerobic exercise would be able to promote different changes in the cardiac autonomic modulation of healthy elderly people and elderly stroke patients. Therefore, the aim of this study is to compare cardiac autonomic modulation in elderly stroke patients and healthy controls before, during, and after an acute session of aerobic exercise.

## 2. Methods

Our study conformed to the STROBE [29] guidelines. The study followed a cross-sectional [30] design that assessed the period between May 2012 and April 2013 with the diagnosis of stroke patients, and October 2015 to November 2019 for healthy elderly controls.

### 2.1. Study Population and Entry Criteria

Sixty elderly people were included in this analysis. The elderly were divided into two groups: the control group (CG—30 healthy elderly individuals) had as inclusion criteria being over 65 years old and not having any cardiovascular or metabolic dysfunction, while the inclusion criteria for the stroke group (SG—30 patients) were having had a stroke diagnosis evidenced by an imaging exam with a medical report, duration of the lesion over one year, and being over 65 years old. All of the patients were clinically evaluated by a cardiologist and underwent a clinical exercise test prior to the experimental protocol. Patients were to walk on the treadmill without any adaptation apparatus; therefore, they had to support the unaffected upper limb on the treadmill and begin the exercise. Patients with other neurological diseases, heart diseases, diabetes mellitus, smokers, drinkers who used beta blockers or antiarrhythmic medication, and those who did not present medical authorization and prescription for physical activity were not included [12].

Elderly people who did not obtain medical clearance to exercise on a treadmill were excluded from both groups; in addition, elderly people who did not adapt to the proposed exercises, or who did not reach the training heart rate proposed by the research, were excluded. Patients who were unable to maintain a good balance were excluded.

### 2.2. Ethics Statement

All subjects gave their written informed consent for this study, which was approved by the Research Ethics Committee (CAEE: 43334714.6.0000.5429, permit number 1.017.631). The Declaration of Helsinki, with its ethical guidelines for medical research on humans, was observed.

### 2.3. Study Design and Bias

To address potential sources of bias, we performed all protocols under the same laboratory conditions. Data were collected in a rehabilitation clinic under controlled temperature (21–25 °C) and humidity (50–60%). Individuals were requested to abstain from caffeine, alcohol, carbonated beverages, chocolate (12 h), and exercise (24 h). Data were collected between 1 p.m. and 3 p.m. to minimize the interference of circadian rhythms. All procedures necessary for the data collection were explained to the individuals, and the subjects were instructed to remain at rest during the data collection [12].

#### 2.3.1. Initial Assessment and Experimental Protocol

Both groups were classified as active according to the International Physical Activity Questionnaire (IPAQ) [31]. The SG was characterized with three tests: the Mini-Mental State Examination (MMSE)—cognitive functions [32], the Fugl-Meyer Assessment—physical performance through aspects of motor control [33], and the Orpington Prognostic Scale [34]—assessment of stroke severity.

The International Physical Activity Questionnaire (IPAQ) was developed to measure health-related physical activity (PA) in populations. Possible categories are: “sedentary” (does not perform any physical activity for at least 10 continuous min during the week); “insufficiently active” (individuals who practice physical activities for at least 10 continuous min per week, but not enough to be classified as active), and “active” or “very active” (comply with weekly exercise time recommendations). The IPAQ is validated for stroke patients [31].

The MMSE is a test used to assess cognitive function because it is fast; it is divided into domains and two steps: the first seeks to assess orientation, memory, and attention, while the second analyzes specific skills such as naming and comprehension (30 points in total). The higher a patient’s score, the better their cognitive performance. A patient who scores more than 25 points is considered normal. Mild cognitive loss is suspected when the score is between 21 and 24 points; moderate, between 10 and 20; and severe, less than or equal to 9 [32].

The Fugl-Meyer Assessment provides conditions to score functional activities; it is divided into five domains: motor function, sensitivity, balance, range of motion, and pain. The domain of motor function includes measurement of movement, coordination, and reflex activity of the shoulder, elbow, wrist, hand, hip, and ankle, totaling 100 points, with 66 referring to the upper extremities and 34 referring to the lower extremities. Depending on the total score, the patient can be classified as having severe, moderate, or mild impairment [33].

The Orpington Prognostic Scale assesses the severity of the stroke, with values up to 3 points representing “good prognosis”, 3 to 5 points referring to a “moderate prognosis”, and above 5 points indicating a “poor prognosis” [34].

After the initial evaluation, the heart monitor strap was placed on each subject’s thorax over the distal third of the sternum. Measurements were subsequently obtained at rest before starting the exercise in the supine position for 10 min, during incremental aerobic exercise for 30 min, and for a further 30 min in supine position post-exercise (recovery period). The stipulated training load for both groups was 60–70% of the maximum HR (HRmax). Based on the study of Tanaka et al. [35], the HRmax calculated was HRmax = 207 − 0.7 × age.

After 10 min in supine position, the subjects were advised to walk on a treadmill (RT250pro, Movement, Sao Paulo, Brazil) for aerobic exercise, divided into five min of warm-up and 25 min of exercise (70% of the heart rate reserve) [35,36]. Blood pressure, respiratory rate, and partial oxygen saturation were measured every five minutes.

The criteria for initiation or interruption of the exercise were: less than or equal to 120 mmHg DBP, SBP above 200 mmHg, chest pain, dyspnea disproportionate to exertion, lack of motor coordination, and fatigue.

#### 2.3.2. Assessment of HRV

Heart rate was recorded beat-to-beat via an HR monitor with a sampling frequency of 1000 Hz (S810i, Polar Electro Oy, Kempele, Finland) and, from these records, time series were generated and used for HRV analysis [15].

HRV was analyzed in the time and frequency domains. The time-domain measurements used were SDNN (standard deviation of normal intervals), RMSSD (root mean square of successive differences and the normal-to-normal R–R), and pNN50 (intervals that differ by more than 50 ms). SDNN is a global index encompassing both the sympathetic and parasympathetic nervous systems. The RMSSD and pNN50 are used as markers of cardiac parasympathetic modulation [13].

The triangular index (RRtri), triangular interpolation of RR intervals (TINN), and Poincaré plot allow the analysis of RR intervals via geometric patterns and approaches to assess HRV. The RRtri and the TINN express the overall variability of RR intervals [13].

The Poincaré plot allows each RR interval to be plotted against the previous interval, and for quantitative analysis of the graph, the following indices were calculated: SD1 (standard deviation of instantaneous beat-to-beat variability), characterized as a parasympathetic cardiac modulation marker; SD2 (standard deviation of long-term continuous RR intervals), characterized as marker of the parasympathetic and sympathetic modulation; and the SD1:SD2 ratio, which may be used during physical exercise as an indicator of sympathetic modulation [13].

The main spectral components of the total power (TP) or variance in RR intervals are the low frequency (LF) and the high frequency (HF). The TP of HRV is an estimation of the global activity of the autonomic nervous system. For the frequency domain, we calculated an LF (0.04–0.15 Hz) power, which has been linked to sympathetic modulation of the heart, HF (0.15–0.40 Hz) power, which has been linked to vagal activity, and cardiac sympathetic–vagal balance, which may be represented by the LF/HF ratio [13]. We calculated the spectral analysis using a fast Fourier transform algorithm. We selected stable series during exercise.

After acquisition, the signals were transferred to the Polar Precision Performance SW software (version 4.01.029), corrected for ectopic beats (digital filtering software), and only series with more than 95% sinus rhythm were included. Then, time-series data were extracted into a “txt” format and processed using the Kubios HRV analysis software (MATLAB, version 2 beta, Kuopio, Finland).

#### 2.3.3. Outcome Measures

The primary outcome was to compare the cardiac autonomic modulation of elderly stroke patients with that of the general population. The secondary outcome was measured in rest—10 min of rest in supine position; exercise—the 20 min of peak exercise (removing the first and the last 5 min of the 30 min exercise); and recovery—30 min in supine position post-exercise. The post-exercise period was divided into R1, the first 5 min of recovery; R2, halfway through the recovery time—the period between the 12th and 17th minutes of recovery—and R3, the final 5 min of recovery. At all times, the RR value exceeded 256 beats.

#### 2.3.4. Study Size

The sample size was selected based on a pilot test, wherein the online software provided by the website www.lee.dante.br was used, taking into consideration the RMSSD index as a variable. The significant difference in magnitude assumed was 14.11 ms, with a standard deviation of 12.8 ms; based on an alpha risk of 5% and beta risk of 80%, the sample size determined was a minimum of 13 individuals per group.

### 2.4. Data Analysis

The results are reported as the mean ± standard deviation, and all statistical analysis was conducted at a 95% level of significance. Baseline characteristics were compared between the 2 groups using 2-sample *t*-tests for continuous variables and Fisher’s exact test for categorical variables. The dependent variables were submitted to a 2 (group: stroke, control) by 5 (moments: rest, exercise, R1, R2, R3) ANOVA with repeated measures on the last factor. The Shapiro–Wilk test was used to verify data normality. Post hoc comparisons were carried out using Tukey’s HSD (honest significant difference) test (*p* < 0.05). The software package used was SPSS 20.0 (Chicago, IL, USA).

## 3. Results

### 3.1. Study Population

The characteristics of the participants are described in Table 1. There were no significant differences in age, gender, race, weight, or blood pressure. Patients from a neurology outpatient clinic were recruited into the stroke group. Of a total of 45 stroke patients, 30 were selected (5 did not want to participate in the study due to difficulty walking to the exercise location, and 10 did not meet the eligibility criteria—8 used beta-blockers and 2 used antiarrhythmic drugs). For the control group, elderly members of the general population who volunteered for an exercise program were recruited from the community. These elderly people were called for initial assessment, and if they agreed to participate in the study they were included. A total of 30 elderly people were initially recruited, but 6 had diabetes mellitus and systemic arterial hypertension; after this pre-selection, 10 more patients were recruited, and 4 did not want to participate in the study.

### 3.2. Time Domain

The main effects for moments were present in all time-domain indices: the comparison for rest with exercise, R1, R2, and R3 presented significant differences (Table 2). The main effect for groups was found only for the mean RR and mean HR indices, where the mean RR was higher and the mean HR was lower in the SG. There were interactions between groups and moments for Mean RR and SDNN. The post hoc test results are displayed in Table 2.

### 3.3. Frequency Domain

The main effects for moments were also present in all frequency-domain indices (Table 3). The main effects for groups were found for the HF (ms^2^) and LF/HF ratio, where the HF (ms^2^) was higher and the LF/HF ratio was lower in the SG. There were interactions between groups and moments for LF (ms^2^), LF (nu), and LF/HF ratio. The post hoc test results are shown in Table 3.

### 3.4. Geometrical

Once more, the main effects for moments were present in all geometrical indices (Table 4). The main effect for groups was found only for SD2, which was lower in the SG. There were interactions between groups and moments for TINN and SD2. The post hoc test results are shown in Table 4.

## 4. Discussion

HRV is a physiological phenomenon that is measured by the variation in the beat-to-beat interval; it can be a useful tool in the assessment of the autonomic nervous system, and can indicate the heart’s ability to respond to various physiological and environmental stimuli, such as physical exercise. Our study demonstrated that, at rest, the indices that represent the sympathovagal balance (LF/HF ratio) and global variability (SD2) were lower in the stroke group than in controls, which may indicate reduced variability.

Reduced variability was found in both elderly and stroke patients. A study by Shibasaki et al. [37], which investigated the relationship between the physical function, mortality, and autonomic nervous activity measured by HRV of the elderly in long-term care, found that LF/HF was significantly decreased in the long-term-care elderly compared with control elderly after adjustment for covariates, and that decrease in LF/HF was an independent risk factor for mortality. Our data presented low values of LF/HF ratio and SD2 in the SG compared to controls.

Conversely, Katz-Leurer and Shochina investigated whether post-stroke autonomic impairment modifies the influence of early aerobic training. Early aerobic exercise significantly improved endurance and walking performance in stroke patients, but no modification effect was found between HRV parameters and exercise on those parameters. It is more appropriate to report LF and HF in normalized units in response to exercise, which is considered a marker of sympathovagal balance [38]. An increase in LF/HF ratio suggests a shift in autonomic activity to sympathetic predominance; this was corroborated by our results, because there was an elevation in LF (ms^2^ and nu) during exercise [39,40].

Additionally, in the frequency domain, Xiong et al. evaluated 77 ischemic stroke patients after stroke and 37 elderly controls. Central autonomic impairment is frequent in ischemic stroke at the acute or chronic stages [10]. The LF was significantly reduced in stroke patients, who also showed impairment in all parasympathetic tests in comparison with controls. The authors concluded that stroke patients showed a parasympathetic cardiac deficit. Although the behavior of LF, in exercise, was the same in both groups, the values after 30 min remained lower than the rest in the stroke group, which may be indicative of autonomic impairment in these individuals.

Galetta et al. investigated the effects of physical activity on HRV and carotid intima–media thickness (IMT) in elderly subjects. Elderly athletes showed evidence of increased vagal activity compared to sedentary subjects [22]. Moreover, athletes showed lower IMT than control subjects. In the whole population, SDNN was inversely related to IMT, while LF/HF ratio was positively related to IMT. In our study, there was a reduction in the vagal indices during exercise in both groups, and there were differences between the groups in terms of returning to baseline values. Soares-Miranda et al. evaluated the association of increased activity levels with HRV measures in older adults. Leisure-time activity and walking distance were significantly related to SDNN, whereas walking pace was inversely related to the Poincaré ratio (SD1/SD2) [7].

With regard to the Poincaré plot and the other geometric indices, in this study, there was a reduction in all indices during exercise in both groups, and there were differences between the groups in terms of returning to baseline values. TINN and SD2—indices of overall autonomic modulation—did not return to their rest values. Acute autonomic responses to exercise have been previously documented [41]. Francica et al. evaluated 14 post-stroke women and 10 general population control women subjected to exercise on a cycle ergometer (10 W · min^−1^, with increments of 0.3 kp every minute, until reaching 80% of maximal HR). As a main finding, the authors concluded that women with chronic stroke had negative changes in vagal regulation of HR [42].

Taken together, our findings indicate that (1) at rest, the stroke group presents a reduced variability; (2) both stroke patients and controls, during exercise, show elevation of sympathetic modulation; and (3) the stroke group does not return to baseline after the 30 min of recovery. Although HRV reference values are still being discussed, some studies have tried to make data for comparison possible. Our results, when compared with the normality values from the studies by Sammito and Böckelmann [14,43] and Dantas et al. [44], showed higher values of SDNN and lower values of pNN50, rMMSD, LF, HF, and LF/HF when compared to same age group.

These findings are clinically important because they point to a probable imbalance in autonomic regulation in these individuals. Some authors have confirmed that the regular physical activity represents a strategy to counter age-related impairments of cardiac autonomic activity in stroke patients [28,45]. Associations between leisure-time activity and walking distance appear to be linear, suggesting that any physical activity is better than none, and more is better [9].

Wanderley et al. assessed the effectiveness of different training protocols on autonomic function and systemic inflammation in elderly adults [11]. There was a significant change in HRV, inflammatory biomarkers, and six-minute walking distance in response to aerobic training. Aerobic training comprised 30 min of aerobic exercise, with the intensity being calculated using 50–80% of HR reserve. The present study worked with 70% of HR reserve; the aerobic intensity of stepping training may improve walking ability in post-stroke patients. Post-stroke patients trained with 70–80% of HR reserve, and significantly improved their six-minute walking distance after training. Although our study did not train these patients, the same training intensity was used.

Yperzeele et al. performed a meta-analysis reporting associations between heart rate variability and baroreceptor sensitivity; however, the interpretability of most studies was limited due to small samples. Our study compared 30 stroke patients with a selected group of older adults without cardiovascular dysfunction or metabolic disorders, which may provide interesting information for the investigation of parameters of autonomic nervous system dysfunction [16].

Our study has some limitations. The inclusion of patients with stroke without beta-blockers limits the external validation of this research due to the widespread use of this drug by this population; in addition, it was not determined in the initial anamnesis whether the patients with stroke used other medications for hypertension. Despite not being the objective of this study, another limitation was the non-application of the MMSE in the control group; thus, there was no possibility of comparing the cognitive status of the elderly. Our stroke patients were chronic, so our results cannot be extrapolated to acute and subacute patients.

Rehabilitation services provide the management of neuromuscular and musculoskeletal disorders that alter functional status. Major weaknesses are described in the implementation of public policies for the rehabilitation care of stroke patients. Public health services must be prepared and integrated into public policies on disability. 

## 5. Conclusions

In conclusion, the present study demonstrates that elderly stroke patients have reduced variability at rest, sympathetic predominance during exercise, and do not return to baseline values after 30 min of recovery, with similar responses found in the group of healthy elderly people. According to the WHO, rehabilitation consists of measures that help people with disabilities to maintain optimal functionality and should be a strategy for equalizing opportunities and social integration. Monitoring the autonomic nervous system can be a cheap and noninvasive tool to help prescribe exercises for this population. Longitudinal studies with stroke patients can elucidate the role of HRV in the prescription and safety of aerobic exercise and, perhaps, provide subsidies to promote an improvement in the quality of rehabilitation provision.

## Figures and Tables

**Table 1 ijerph-18-11460-t001:** Baseline characteristics of the study groups.

Characteristics	Control Group (*n* = 30)	Stroke Group (*n* = 30)	*p*
Age (mean ± SD, years)	67 ± 4	69 ± 3	0.096
Sex (% male)	80	70	0.093
Caucasians (%)	85	90	0.532
Body weight (kg)	79 ± 6	71 ± 9	0.237
Resting SBP (mean + SD, mmHg)	120 ± 5	125 ± 3	0.075
Resting DBP (mean + SD, mmHg)	80 ± 4	82 ± 3	0.090
Type of injury (%, ischemic)	-	49	-
Hemisphere of injury (%, left)	-	90	-
Time of injury (years)	-	7 ± 3	-
MMSE	-	19.3 ± 4	-
FMul	-	21.2 ± 6	-
FMII	-	20.4 ± 4	-
FM balance	-	9.6 ± 1	-
FM sensibility	-	20.6 ± 3	-
FM ROM	-	38.5 ± 4	-
FM pain	-	36.6 ± 2	-
FM total	-	170.5 ± 9	-
Orpington	-	2.9 ± 1	-
IPAQ	Active	Active	-

*p*: *p*-value; SD: standard deviation; SBP: systolic blood pressure; DBP: diastolic blood pressure; MMSE: Mini-Mental State Examination; FMul: Fugl-Meyer Assessment for the upper limbs; FMll: Fugl-Meyer Assessment for the lower limbs; FM: Fugl-Meyer Assessment; ROM: range of motion; IPAQ: International Physical Activity Questionnaire.

**Table 2 ijerph-18-11460-t002:** Analysis of HRV in the time domain at rest, aerobic exercise, and recovery (control group vs. stroke group).

Variables	Control Group					Stroke					(df) F ŋ^2^ (*p*)		
	Rest Mean (SD)	Exercise Mean (SD)	R1 Mean (SD)	R2 Mean (SD)	R3 Mean (SD)	Rest Mean (SD)	Exercise Mean (SD)	R1 Mean (SD)	R2 Mean (SD)	R3 Mean (SD)	Main Effect: Moment	Main Effect: Group	Interaction: Moment vs. Group
MeanRR (ms)	767.43 (110.98)	542.31 (59.54)	711.26 (96.29)	738.97 (110.97)	765.00 (105.52)	859.51 (107.09)	578.52 (89.78)	755.35 (95.85)	792.71 (97.08)	808.83 (100.13)	(4.312) 321.1 0.80 (<0.001) *	(1.78) 7.6 0.09 (0.007)	(4.312) 3.9 0.05 (0.013) ^†^
SDNN (ms)	56.60 (23.29)	30.25 (15.36)	33.34 (17.78)	32.13 (15.41)	33.81 (17.96)	42.56 (18.36)	27.78 (17.01)	30.26 (10.66)	28.08 (9.71)	33.81 (12.17)	(4.312) 32.8 0.30 (<0.001) *	(1.78) 3.6 0.04 (0.062)	(4.312) 32.8 0.30 (<0.001) ^‡^
MeanHR (bpm)	80.27 (10.74)	112.60 (13.27)	84.18 (8.69)	81.77 (9.45)	79.72 (9.60)	71.13 (8.54)	106.69 (17.13)	80.82 (9.73)	76.93 (9.16)	75.62 (9.17)	(4.312) 359.8 0.82 (<0.001) *	(1.78) 7.2 0.08 (0.009)	(4.312) 2.3 0.03 (0.108)
rMSSD (ms)	23.53 (14.63)	8.75 (4.99)	19.33 (12.56)	20.05 (15.59)	19.27 (12.64)	24.28 (10.43)	8.11 (3.62)	22.11 (14.22)	19.80 (11.54)	21.98 (10.71)	(4.312) 41.6 0.35 (<0.001) *	(1.78) 0.3 0.00 (0.601)	(4.312) 0.8 0.01 (0.518)
pNN50 (%)	6.44 (5.27)	0.53 (0.69)	2.41 (2.74)	2.78 (3.34)	4.39 (5.93)	5.76 (4.06)	0.25 (0.26)	2.00 (2.39)	2.34 (2.86)	2.38 (2.69)	(4.312) 46.3 0.37 (<0.001) *	(1.78) 2.0 0.02 (0.166)	(4.312) 1.4 0.02 (0.260)

R1: recovery period 1—first 5 min of the recovery period; R2: recovery period 2—between the 10th and 15th minutes of the recovery period; R3: recovery period 3—between the 25th and 30th minutes of the recovery period; MeanRR: average RR interval; ms: milliseconds; SDNN: standard deviation of all normal RR intervals recorded at an interval of time; bpm: beats per minute; MeanHR: average heart rate; rMSSD: root mean square of the square of differences between adjacent normal RR intervals in an interval of time; pNN50: percentage of adjacent RR intervals with a difference in duration greater than 50 ms; %: percentage; SD: standard deviation; *: post hoc test found differences when comparing rest with all other moments; ^†^: post hoc test found differences when comparing rest with all other moments, except between rest and R3 for the control group; ^‡^: post hoc test found differences when comparing rest with all other moments, for both groups.

**Table 3 ijerph-18-11460-t003:** Analysis of HRV in the frequency domain at rest, aerobic exercise, and recovery (control group vs. stroke group).

Variables	Control Group					Stroke					(df)F ŋ^2^ (*p*)		
	Rest Mean (SD)	Exercise Mean (SD)	R1 Mean (SD)	R2 Mean (SD)	R3 Mean (SD)	Rest Mean (SD)	Exercise Mean (SD)	R1 Mean (SD)	R2 Mean (SD)	R3 Mean (SD)	Main Effect: Moment	Main Effect: Group	Interaction: Moment vs. Group
LF (ms^2^)	341.89 (257.03)	444.50 (135.79)	169.70 (107.99)	391.75 (286.83)	365.92 (265.84)	373.28 (228.38)	437.72 (132.92)	132.23 (66.94)	226.04 (135.67)	267.85 (227.40)	(4.312) 59.6 0.43 (<0.001) *	(1.78) 3.7 0.04 (0.059)	(4.312) 5.0 0.06 (0.002) ^‖^
LF (nu)	69.49 (19.02)	72.93 (14.56)	75.04 (17.20)	73.21 (15.31)	72.83 (17.09)	68.77 (16.01)	78.25 (10.90)	64.95 (16.42)	69.93 (13.59)	62.48 (20.33)	(4.312) 4.1 0.05 (0.004) ^†^	(1.78) 2.5 0.03 (0.120)	(4.312) 4.8 0.06 (0.001) ^#^
HF (ms^2^)	177.98 (145.62)	9.68 (6.07)	85.90 (67.31)	108.68 (56.39)	95.20 (68.03)	170.05 (102.13)	7.63 (5.03)	87.02 (51.05)	108.45 (41.66)	123.34 (66.41)	(4.312) 78.9 0.50 (<0.001) ^‡^	(1.78) 314.5 0.80 (<0.001)	(4.312) 1.1 0.01 (0.334)
HF (nu)	30.08 (17.33)	25.92 (12.03)	29.76 (15.92)	27.75 (17.86)	27.79 (14.77)	31.29 (16.02)	21.87 (10.09)	35.30 (16.10)	30.07 (13.59)	32.88 (18.02)	(4.312) 5.4 0.07 (<0.001) ^†^	(1.78) 0.7 0.01 (0.393)	(4.312) 1.9 0.02 (0.116)
LF/HF (-)	4.16 (2.94)	3.90 (2.46)	4.06 (3.06)	5.74 (4.69)	4.44 (3.34)	2.98 (1.52)	4.33 (2.00)	2.60 (1.86)	3.09 (2.03)	3.46 (2.62)	(4.312) 2.8 0.03 (0.040) ^§^	(1.78) 7.8 0.09 (0.007)	(4.312) 4.5 0.05 (0.004)

R1: recovery period 1—first 5 min of the recovery period; R2: recovery period 2—between the 10th and 15th minutes of the recovery period; R3: recovery period 3—between the 25th and 30th minutes of the recovery period; LF: low-frequency component; ms^2^: milliseconds squared; HF: high-frequency component; LF/HF: ratio between the low-frequency and high-frequency components; SD: standard deviation; *: post hoc test found differences when comparing rest with exercise and with R1; ^†^ post hoc test found differences when comparing Rest with Exercise; ^‡^: post hoc test found differences when comparing rest with all other moments; ^§^: post hoc test found differences when comparing rest with R2; ^‖^: differences found when comparing rest with all other moments, except for rest versus R2 and rest versus R3 for the control group; ^#^: differences found only for rest versus exercise and rest versus R3 for the stroke group.

**Table 4 ijerph-18-11460-t004:** Analysis of geometrical indices at rest, aerobic exercise, and recovery (control group vs. stroke group).

Variables	Control Group					Stroke					(df)F ŋ^2^ (*p*)		
	Rest Mean (SD)	Exercise Mean (SD)	R1 Mean (SD)	R2 Mean (SD)	R3 Mean (SD)	Rest Mean (SD)	Exercise Mean (SD)	R1 Mean (SD)	R2 Mean (SD)	R3 Mean (SD)	Main Effect: Moment	Main Effect: Group	Interaction: Moment vs. Group
RRTri (-)	9.97 (4.39)	5.70 (2.15)	7.81 (3.07)	7.86 (3.43)	7.61 (3.08)	8.98 (3.15)	4.91 (1.67)	7.48 (2.54)	7.08 (2.19)	7.85 (2.62)	(4.312) 29.2 0.27 (<0.001) *	(1.78) 1.5 0.02 (0.224)	(4.312) 0.8 0.01 (0.501)
TINN (ms)	220.03 (87.35)	141.67 (79.44)	144.74 (76.23)	135.91 (66.93)	137.74 (60.77)	161.40 (71.40)	120.97 (70.97)	130.44 (46.78)	122.62 (49.65)	146.95 (63.16)	(4.312) 16.0 0.17 (<0.001) *	(1.78) 3.7 0.04 (0.059)	(4.312) 3.8 0.05 (0.014) ^§^
SD1 (ms)	15.65 (7.96)	5.76 (3.02)	12.94 (9.40)	13.49 (9.13)	15.15 (12.67)	16.85 (7.41)	5.60 (2.36)	14.51 (8.94)	13.56 (7.21)	15.12 (7.24)	(4.312) 43.0 0.36 (<0.001) *	(1.78) 0.1 0.00 (0.705)	(4.312) 0.4 0.01 (0.773)
SD2 (ms)	69.26 (23.58)	41.64 (20.53)	40.78 (14.54)	43.02 (20.99)	43.50 (19.38)	53.22 (18.12)	37.23 (21.59)	38.93 (14.93)	36.47 (13.10)	43.91 (15.49)	(4.312) 29.0 0.27 (<0.001) ^†^	(1.78) 4.3 0.05 (0.041)	(4.312) 3.3 0.04 (0.022) ^‡^
SD1/SD2	0.23 (0.16)	0.17 (0.12)	0.31 (0.24)	0.45 (0.72)	0.36 (0.23)	0.33 (0.19)	0.22 (0.17)	0.41 (0.28)	0.40 (0.25)	0.37 (0.21)	(4.312) 8.1 0.09 (<0.001) ^†^	(1.78) 1.1 0.01 (0.301)	(4.312) 1.0 0.01 (0.383)

R1: recovery period 1—first 5 min of the recovery period; R2: recovery period 2—between the 10th and 15th minutes of the recovery period; R3: recovery period 3—between the 25th and 30th minutes of the recovery period; RRtri: triangular index; TINN: triangular interpolation histogram of RR intervals; SD1: standard deviation of the variability instantaneous beat-to-beat or dispersion of points perpendicular to the line of identity; SD2: standard deviation of long-term continuous RR intervals or dispersion of points along the line of identity; SD1/SD2: ratio scatter of points perpendicular to the dispersion of dots along the line of identity; *: post hoc test found differences when comparing rest with all other moments, except for R3; ^†^: post hoc test found differences when comparing rest with all other moments; ^‡^: differences found when comparing rest with all other moments, for both groups; ^§^: post hoc test found differences when comparing rest with all other moments, except between rest and R3 for the stroke group.

## Data Availability

The data presented in this study are available on request from the corresponding author.

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
