# Peer review of "The Use of Cardiac Autonomic Responses to Aerobic Exercise in Elderly Stroke Patients: Functional Rehabilitation as a Public Health Policy"

_ijerph, 2021, doi:10.3390/ijerph182111460_

Round 1
Reviewer 1 Report
Thank you for submitting the manuscript.
The aim of this study was to evaluate the changes in heart rate variability of elderly healthy subjects and stroke patients. The results show reduced HRV in elderly stroke group. This study raises some other questions and comments for me. In particular, I think the inadequate description of stroke patients is especially relevant. It can be generally said that about 1/3 of the stroke patients have no residuals at all. The lack of interpretation and comparability with the control group limited the assessment of HRV.
General comments
- The reference list does not correspond to one of the two recommended reference styles. The author guidelines should be read and the reference list should be adjusted. Use [] for citation in the text.
- Under all tables, explain abbreviations, signs and the asterisks when listing statistics. For example, what does § mean in table 4? Please check all tables regarding other characters.
- All variables and R3, main effect etc. should be marked in bold. Please check all tables.
Introduction
- The reader is not introduced to the method of HRV at all. It remains unclear what HRV means. How does a normal stress reaction proceed in healthy people; in this example under physical stress? What is known about this in stroke patients? A reader who is not familiar with HRV cannot do anything at all with the meaning of the data. Please add to that. Without this information, the meaning of your work cannot be explained to every reader.
- Yperzeele L, van Hooff R-J, Nagels G et al. (2015) Heart rate variability and baroreceptor sensitivity in acute stroke: a systematic review. Int J Stroke 10: 796–800. https://doi.org/10.1111/ijs.12573
- What HRV guidelines were used? Which reference values for the HRV parameters were used? Which confounders can influence HRV?
- I recommend common guidelines such as:
- Task Force of the European Society of Cardiology and the North American Society of Pacing and Electrophysiology (1996): Heart rate variability. Standards of measurement, physiological interpretation, and clinical use. 0041. In: European heart journal 17 (3), S. 354–38
- Sammito, Stefan; Thielmann, Beatrice; Seibt, Reingard; Klussmann, André; Weippert, Matthias; Böckelmann, Irina (2015): Guideline for the application of heart rate and heart rate variability in occupational medicine and occupational science. 0651. In: ASU International. DOI: 10.17147/ASUI.2015-06-09-03.
- Your national guideline, if applicable.
Methods
- The citation for the reference to the STROBE guidelines is missing and should be added.
- STROBE guidelines also include the timeline of investigations. No information is provided on this. Please add the time period of the examinations.
- Study population: I can't believe that the stroke patients didn't take any medications that affect HRV. What about hypertension and antihypertensive drugs? Cardiac arrhythmias and hypertension are common causes in stroke patients. I request a comment on hypertension and drugs in both groups because hypertension decreases HRV. Was it comparable? If it was not collected, it should be included as a limitation of the study.
- Shouldn’t the duration of stroke and elevation be more detailed? What is the functional status of stroke patients? In general, it can be said that 1/3 of the stroke patients have no residuals anymore.
- How should the data from Table 1 (e.g., MMSE) be interpreted? Are the data good or poor? I can well imagine that a greater loss of function also means more physical stress and thus can influence the HRV parameter.
- Was the training load of both groups the same?
- Ethics statement: It should be added that the Declaration of Helsinki with its ethical guidelines of medical research on humans was observed.
- Initial assessment: An interpretation should be presented to evaluate the values of Table 1. Why are these missing for the control group? Please add to establish comparability. If these data are missing, it should be listed as a limitation of the study. In principle, it is conceivable that the stroke patients no longer have residuals and thus have good function similar to the control group.
- Assessment of HRV: The comparability of HRV data from different recording lengths is methodologically objectionable and not meaningful. It should be corrected.
- The analysis method in frequency domain parameters should be supplemented (e.g. Fast Fourier Transform, Autoregressive Model, Trigonometric Regressive Spectral Analysis). FFT analysis requires stationarity and is not present in the exercises.
Discussion
- The discussion should be revised. E.g. the 1st paragraph can be moved to the introduction (see comment 3).
- Lines 46-48: Which reference values are referred to here?
- What is the new knowledge in addition to the systematic review?:
- Yperzeele L, van Hooff R-J, Nagels G et al. (2015) Heart rate variability and baroreceptor sensitivity in acute stroke: a systematic review. Int J Stroke 10: 796–800. https://doi.org/10.1111/ijs.12573
- The discussion should end with a conclusion: what is the clinical relevance of the study? Which measures are recommended for the rehabilitation of patients or for further research?
- Study limitations should be listed and discussed. Currently, some limitations are found, e.g., different duration of HRV recording between phases, lack of specification of the analysis method for the frequency domain, insufficient description of the functional status of stroke patients, period of stroke and study period, hypertension history of stroke patients, and intake of antihypertensive medication.
Author Response
To: Aerobic exercise in elderly people with stroke as a public health policy
Manuscript ID: ijerph-1385475
Type of manuscript: Article
Authors: Rodrigo Daminello Raimundo, Juliana Zangirolami-Raimundo, Claudio Leone, Tatiana Dias de Carvalho, Talita Dias da Silva, Italla Maria Pinheiro Bezerra, Alvaro Dantas de Almeida-Junior, Vitor Engracia Valenti, Luiz Carlos de Abreu
Dear Dr.
We very much appreciate you for your highly constructive reviews to our submission. We have revised the material to eliminate the issues raised. The added or modified words, phrases, and sentences are in red. We hope the present version can be accepted.
Review 1
Thank you for submitting the manuscript.
The aim of this study was to evaluate the changes in heart rate variability of elderly healthy subjects and stroke patients. The results show reduced HRV in elderly stroke group. This study raises some other questions and comments for me. In particular, I think the inadequate description of stroke patients is especially relevant. It can be generally said that about 1/3 of the stroke patients have no residuals at all. The lack of interpretation and comparability with the control group limited the assessment of HRV.
Answer:
Thank you for the attention on our manuscript. As suggested, we have re-written the manuscript in order to improve it. We really appreciate the kind suggestions and recommendations of the reviewers, which significantly helped us to improve the manuscript. We hope the present version can be accepted for publication on this very important journal.
General comments
- The reference list does not correspond to one of the two recommended reference styles. The author guidelines should be read and the reference list should be adjusted. Use [ ] for citation in the text.
Answer: Reference list and style were adjusted (Mendeley reference manager)
- Under all tables, explain abbreviations, signs and the asterisks when listing statistics. For example, what does § mean in table 4? Please check all tables regarding other characters.
Answer: Thank you for your comment. All abbreviations, signs and asterisks have been revised and inserted into the caption
- All variables and R3, main effect etc. should be marked in bold. Please check all tables.
Answer: Thank you for your comment. All variables and R3, main effect were marked in bold
Introduction
- The reader is not introduced to the method of HRV at all. It remains unclear what HRV means. How does a normal stress reaction proceed in healthy people; in this example under physical stress? What is known about this in stroke patients? A reader who is not familiar with HRV cannot do anything at all with the meaning of the data. Please add to that. Without this information, the meaning of your work cannot be explained to every reader.
Answer: Thank you for taking the time to send us your feedback. The introduction was rewritten according to reviewers' comments. All changes are in red.
- What HRV guidelines were used? Which reference values for the HRV parameters were used? Which confounders can influence HRV?
- I recommend common guidelines such as:
- Task Force of the European Society of Cardiology and the North American Society of Pacing and Electrophysiology (1996): Heart rate variability. Standards of measurement, physiological interpretation, and clinical use. 0041. In: European heart journal 17 (3), S. 354–38
- Sammito, Stefan; Thielmann, Beatrice; Seibt, Reingard; Klussmann, André; Weippert, Matthias; Böckelmann, Irina (2015): Guideline for the application of heart rate and heart rate variability in occupational medicine and occupational science. 0651. In: ASU International. DOI: 10.17147/ASUI.2015-06-09-03.
Answer:
Thank you for your comment. References cited by reviewers were used to define HRV guidelines. Reference values cited by Sammito et al, as well as confounding factors, were incorporated into the discussion.
- Your national guideline, if applicable.
Answer: Yes, these are the references used in Brazil and that were used in the research
Methods
- The citation for the reference to the STROBE guidelines is missing and should be added.
Answer: Thanks for remembering that. The reference was added
von Elm E, Altman DG, Egger M, Pocock SJ, Gøtzsche PC, Vandenbroucke JP; STROBE Initiative. The Strengthening the Reporting of Observational Studies in Epidemiology (STROBE) statement: guidelines for reporting observational studies. Lancet. 2007 Oct 20;370(9596):1453-7. doi: 10.1016/S0140-6736(07)61602-X. PMID: 18064739.
- STROBE guidelines also include the timeline of investigations. No information is provided on this. Please add the time period of the examinations.
Answer: Thank you for your comment. Various information has been added in the method section.
- Study population: I can't believe that the stroke patients didn't take any medications that affect HRV. What about hypertension and antihypertensive drugs? Cardiac arrhythmias and hypertension are common causes in stroke patients. I request a comment on hypertension and drugs in both groups because hypertension decreases HRV. Was it comparable? If it was not collected, it should be included as a limitation of the study.
Answer: Thank you for your comment. The collection of this study took a long time due to the healthy elderly group, it was very difficult to get a number of elderly people who did not use drugs for arrhythmias, hypertension or any other metabolic disease but after 4 years we got these data. As for the stroke group, we did not include patients who used beta-blockers. Other medications for arrhythmia and hypertension were not asked in the initial anmesis, we inserted this in the limitations.The information below was included in the discussion:
“The inclusion of stroke patients without beta-blockers limits the external validation of this work due to the large use of this drug by this population, in addition, it was not collected in the initial anamnesis if the stroke patients used other medications for hypertension”
- Shouldn’t the duration of stroke and elevation be more detailed? What is the functional status of stroke patients? In general, it can be said that 1/3 of the stroke patients have no residuals anymore.
Answer: Thank you for your comment. More details about the assessment instruments have been added. The Fugl-Meyer assessment is widely applied in international experiments aimed at individuals with hemiparesis. Tests provide conditions for scoring functional activities. It is divided into five domains: motor function, sensitivity, balance, range of motion and pain. The domain of motor function includes measurement of movement, coordination and reflex activity of the shoulder, elbow, wrist, hand, hip and ankle, totaling 100 points, with 66 referring to the upper extremity and referring to the lower extremity. Depending on the total score, the patient can be classified as having severe, moderate or mild impairment.
- How should the data from Table 1 (e.g., MMSE) be interpreted? Are the data good or poor? I can well imagine that a greater loss of function also means more physical stress and thus can influence the HRV parameter.
Answer: Thank you for your comment. The MMSE scale was detailed in the method. The MMSE is a test used to assess cognitive function because it is fast. It is divided into domains and divided into two steps. The first seeks to assess orientation, memory and attention and the second analyzes specific skills such as naming and comprehension (30 points in total). The higher a patient's score, the better their cognitive performance. A patient who scores more than 25 points is considered normal. Mild cognitive loss is suspected when the score is between 21 and 24 points, moderate, between 10 and 20, and severe less than or equal to 9. [22]
- Was the training load of both groups the same?
Answer: Thank you for your comment. The information was added. After the initial evaluation the heart monitor strap was placed on each subject’s thorax over the distal third of the sternum. Measurements were subsequently obtained at rest before starting the exercise on the respective bench press for 10 minutes (min), during incremental aerobic exercise for 30 min and more 30 min bench press post exercise (recovery period). The stipulated training load for both groups was 60-70% of the maximum HR (HRmax). Based on Tanaka et al. the HRmax calculated was HRmax = 207 – 0.7 × age.
Tanaka H, Monahan KD, Seals DR: Age-predicted maximal heart rate revisited. J Am Coll Cardiol 2001; 37: 153-156.
- Ethics statement: It should be added that the Declaration of Helsinki with its ethical guidelines of medical research on humans was observed.
Answer: Thanks for remembering that. All subjects gave their written informed consent for this study, which was approved by Research Ethics Committee (CAEE: 43334714.6.0000.5429, permit number 1.017.631). The Declaration of Helsinki with its ethical guidelines of medical research on humans was observed.
- Initial assessment: An interpretation should be presented to evaluate the values of Table 1. Why are these missing for the control group? Please add to establish comparability. If these data are missing, it should be listed as a limitation of the study. In principle, it is conceivable that the stroke patients no longer have residuals and thus have good function similar to the control group.
Answer: Thanks for the comment, but table 1 has no data missing. All possible comparisons have been made. Data as Type Injury (%, ischemic); Hemisphere of injury (%, left) and Time injury (years) cannot be compared as the control group has no neurological damage. Also, functionality scales such as Fugl-Meyer and Orpington severity are not used on uninjured people. Comments from the Mini-Mental State Examination were inserted in the study limitation.
- Assessment of HRV: The comparability of HRV data from different recording lengths is methodologically objectionable and not meaningful. It should be corrected. The analysis method in frequency domain parameters should be supplemented (e.g. Fast Fourier Transform, Autoregressive Model, Trigonometric Regressive Spectral Analysis). FFT analysis requires stationarity and is not present in the exercises.
Answer: Thanks for the comments. The evaluation of HRV performed in this research following the norms of the Task Force of the European Society of Cardiology and the North American Society of Pacing and Electrophysiology (Circulation 1996). The authors of this manuscript (Rodrigo Daminello Raimundo, Vitor Engracia Valenti and Luiz Carlos de Abreu) total more than 200 publications on the subject.
In order to provide further details regarding HRV analysis, we added the following sentence:
Method, 2.3.2, 5th paragraph:
“We calculated the spectral analysis using the Fast Fourier Transform algorithm. We selected stable series during exercise.”
https://pubmed.ncbi.nlm.nih.gov/?term=Valenti+VE&sort=pubdate
https://pubmed.ncbi.nlm.nih.gov/?term=Abreu+LC&sort=pubdate
https://pubmed.ncbi.nlm.nih.gov/?term=Raimundo+RD&sort=pubdate
https://pubmed.ncbi.nlm.nih.gov/?term=Daminello-Raimundo+R&sort=pubdate
Discussion
- The discussion should be revised. E.g. the 1st paragraph can be moved to the introduction (see comment 3).
Answer: Thanks for the comment. The first paragraph was sent for the introduction and complemented by another previous paragraph as "comment 3"
- Lines 46-48: Which reference values are referred to here?
Answer: Thanks for the comment. Reference values in HRV are still questionable, so the option of this research was to use a control group as similar as possible, even so, the references below were inserted and used in the discussion. Although reference values in HRV are still discussed, some studies have tried to make data for comparison possible. Our results showed, when compared with normality values from the researches by Sammito & Böckelmann and Dantas et al higher values of SDNN and lower values of pNN50, rMMSD, LF, HF and LF/HF when compared to same age group.
- Sammito S, Böckelmann I. New reference values of heart rate variability during ordinary daily activity. Heart Rhythm. 2017 Feb;14(2):304-307. doi: 10.1016/j.hrthm.2016.12.016. Epub 2016 Dec 14. PMID: 27986556.
- Dantas EM, Kemp AH, Andreão RV, da Silva VJD, Brunoni AR, Hoshi RA, Bensenor IM, Lotufo PA, Ribeiro ALP, Mill JG. Reference values for short-term resting-state heart rate variability in healthy adults: Results from the Brazilian Longitudinal Study of Adult Health-ELSA-Brasil study. Psychophysiology. 2018 Jun;55(6):e13052. doi: 10.1111/psyp.13052. Epub 2018 Jan 2. PMID: 29292837.
- Sammito S, Böckelmann I. Reference values for time- and frequency-domain heart rate variability measures. Heart Rhythm. 2016 Jun;13(6):1309-16. doi: 10.1016/j.hrthm.2016.02.006. Epub 2016 Feb 12. PMID: 26883166.
- What is the new knowledge in addition to the systematic review?:
- Yperzeele L, van Hooff R-J, Nagels G et al. (2015) Heart rate variability and baroreceptor sensitivity in acute stroke: a systematic review. Int J Stroke 10: 796–800. https://doi.org/10.1111/ijs.12573
Answer: Thanks for the comment. We understand that the review carried out by Yperzeele et al kept doubts due to “varied methodology, small samples, survival selection and exclusion of patients with frequent comorbidities in stroke”. Our study tries to compare stroke patients with a group of elderly people without cardiovascular and metabolic alterations (which is rare). Data collection for this group took a long time due to this fact. We believe that comparing a group with these characteristics is new.
Information from the work of Yperzeele et al was added to the discussion.
- The discussion should end with a conclusion: what is the clinical relevance of the study? Which measures are recommended for the rehabilitation of patients or for further research?
Answer: Thanks for the comment. The last paragraph was rewritten. Rehabilitation services provide the management of neuromuscular and musculoskeletal disorders that alter functional status. Major weaknesses are described in the implementation of public policies for the rehabilitation care of stroke patients. Public health services must be prepared and integrated into public policies on disability. In conclusion, the present study demonstrated that elderly patients with stroke have reduced variability at rest, sympathetic predominance during exercise and do not return to baseline values over the 30 minutes of recovery, with similar responses found in the group of healthy elderly people. According to WHO, rehabilitation are measures that help people with disabilities to maintain optimal functionality and should be a strategy for equalizing opportunities and social integration, monitoring the autonomic nervous system can be a cheap and non-invasive tool to help prescribe exercises for this population. Longitudinal studies with stroke patients can elucidate the role of HRV in the prescription and safety of aerobic exercise and, perhaps, provide subsidies to promote an improvement in the quality of rehabilitation provision.
- Study limitations should be listed and discussed. Currently, some limitations are found, e.g., different duration of HRV recording between phases, lack of specification of the analysis method for the frequency domain, insufficient description of the functional status of stroke patients, period of stroke and study period, hypertension history of stroke patients, and intake of antihypertensive medication.
Answer: Thanks for the comment. A paragraph on study limitation was inserted.
Our study has some limitations. The inclusion of patients with stroke without beta-blockers limits the external validation of this research due to the large use of this drug by this population, in addition, it was not collected in the initial anamnesis if the patients with stroke used other medications for hypertension. Despite not being the objective of the study, another limitation was the non-application of the MMSE in the control group, thus, there is no possibility of comparing the cognitive status of the elderly.
Review 2
Dear Dr.
We very much appreciate you for your highly constructive reviews to our submission. We have revised the material to eliminate the issues raised. The added or modified words, phrases, and sentences are in red. We hope the present version can be accepted.
Comments and Suggestions for Authors
This study was to investigate the differences on autonomic modulation between elderly adults with chronic stroke and healthy controls at rest, during exercise and post exercise. There are my comments.
The title does not appear to reflect the main purpose of this study.
Answer: Thanks for the comment. The title was changed.
The use of cardiac autonomic responses to aerobic exercise in elderly stroke patients: functional rehabilitation as a public health policy
2.Introduction:
- It has been known that aerobic exercise training is beneficial to individuals after stroke, including elderly stroke patients. The justification of this study should be strengthened up. For example, why is it important to explore the autonomic modulation during exercise and post exercise, especially from the perspective of public health policy?
Answer: Thank you for the attention on our manuscript. As suggested, we have re-written the manuscript in order to improve it. We really appreciate the kind suggestions and recommendations of the reviewers, which significantly helped us to improve the manuscript.
The introduction and discussion of the manuscript were rewritten, strengthening the importance of exploring autonomous modulation during exercise and post-exercise from the perspective of public health policy. All changes are in red.
Ø LN73: It is inappropriate to cite Ref 3, 4, and 12. For example, Ref 12 did not involve exercise training.
Answer: Thanks for the comment. References have been changed
Ø LN 91‐94: The function of autonomic nervous system is to help our body to response to external stimuli, such as exercise. Regarding the hypothesis, what do you mean by “an acute bout of aerobic exercise would be able to promote changes in cardiac autonomic modulation of elderly stroke patients”? Please clarify.
Answer: Thanks for the comment, the sentence did not reflect the study hypothesis. The paragraph has been rewritten. Thus, the need for effective stroke rehabilitation is likely to remain an essential part of the stroke care continuum for the foreseeable future and in improving health policies for functional rehabilitation. However, there are few studies comparing an elderly stroke population to healthy elderly control subjects. Our hypothesis is that an acute session of aerobic exercise would be able to promote different changes in the cardiac autonomic modulation of healthy elderly people with elderly people with stroke. Therefore, the aim of this study is to compare cardiac autonomic modulation in elderly stroke patients before, during and after an acute session of aerobic exercise.
- Method:
Ø LN104‐110: Please specifically list your inclusion and exclusion criteria for stroke participants and healthy participants.
Answer: Thank you for your comment. Various information has been added in the method section.
Ø LN126‐127: Please provide the cutoff score for classifying the level of physical activity. In addition, is IPAQ valid for stroke population?
Answer: Thank you for your comment. More details about the assessment instruments have been added. Information about the IPAQ has been inserted into the method. The IPAQ is used for the stroke population as well. The IPAQ is a validated instrument that allows estimating the weekly time spent on physical activities. Possible categories are: "Sedentary" (does not perform any physical activity for at least 10 continuous minutes during the week); "Insufficiently Active" (individuals who practice physical activities for at least 10 continuous minutes per week, but not enough to be classified as active) and "Active" or "Very Active" (comply with weekly exercise time recommendations. The IPAQ is validated for stroke patients
Ruescas-Nicolau MA, Sánchez-Sánchez ML, Cortés-Amador S, Pérez-Alenda S, Arnal-Gómez A, Climent-Toledo A, Carrasco JJ. Validity of the International Physical Activity Questionnaire Long Form for Assessing Physical Activity and Sedentary Behavior in Subjects with Chronic Stroke. Int J Environ Res Public Health. 2021 Apr 29;18(9):4729. doi: 10.3390/ijerph18094729. PMID: 33946690; PMCID: PMC8125179.Ø LN 127‐130: Brief introductions on those assessment tools should be added.
Answer: Thank you for your comment. The method has been rewritten with the necessary information.
Ø LN137‐140: what kind of equation did you use to calculate age‐predicted maximal heart rate?
Answer: Thank you for your comment. The method has been rewritten with the necessary information. After the initial evaluation the heart monitor strap was placed on each subject’s thorax over the distal third of the sternum. Measurements were subsequently obtained at rest before starting the exercise on the respective bench press for 10 minutes (min), during incremental aerobic exercise for 30 min and more 30 min bench press post exercise (recovery period). The stipulated training load for both groups was 60-70% of the maximum HR (HRmax). Based on Tanaka et al. the HRmax calculated was HRmax = 207 – 0.7 × age.
References:
- Sammito, Stefan; Thielmann, Beatrice; Seibt, Reingard; Klussmann, André; Weippert, Matthias; Böckelmann, Irina (2015): Guideline for the application of heart rate and heart rate variability in occupational medicine and occupational science. 0651. In: ASU International. DOI: 10.17147/ASUI.2015-06-09-03.
- Camarda SR, Tebexreni AS, Páfaro CN, Sasai FB, Tambeiro VL, Juliano Y et al. Comparison of maximal heart rate using the prediction equations proposed by Karvonen and Tanaka. Arq Bras Cardiol. 2008;91:311–4.
- Robergs RA, Landwehr R. The surprising history of the Hrmax =“220-age” equation. J Exerc Physiol. 2002;5(2):1–10.
- Tanaka H, Monahan KD, Seals DR. Age—predicted maximal heart revisited. J Am Coll Cardiol. 2001;37:153–6.
Ø LN174‐176: How many data were excluded? Please report this information in the result section.
Answer: Thank you for your comment. The results was rewriten. Patients from a neurology outpatient clinic were recruited into the stroke group. Of a total of 45 stroke patients, 30 were selected (five did not want to participate in the study - difficulty walking to the exercise location and 10 did not meet the eligibility criteria - eight used beta-blockers and two used antiarrhythmic drugs). For the group of general population were re-cruited from the community who volunteered for an exercise program. These elderly people were called for initial assessment and if they ac-cepted to participate in the study they were included. 30 elderly people were initially recruited, but six had diabetes mellitus and / or systemic arterial hypertension, after this pre-selection, 10 more patients were re-cruited and 4 did not want to participate in the study.
Ø LN 180‐188: This paragraph is confusing. Especially, the statement, “The secondary outcome was to compare cardiac autonomic modulation of elderly people with stroke with general population.”, should not this be the main purpose of this study?
Answer: Thanks for the comment. The outcomes are reversed.
The primary outcome was to compare cardiac autonomic modulation of elderly people with stroke with the general population. The secondary outcome was measured in: Rest- 10 min of rest in supine, Exercise- the 20 minutes of peak exercise (removing the first and the last 5 min of 30 min exercise) and Recovery- 30 min in supine post exercise. The post exercise period was divided into (R1) first 5 min of recovery; (R2) half of the recovery time, the period between the twelfth- (12th) and seventeenth- (17th) minute recovery; (R3) final 5 min of recovery. At all times the amount of RR exceeded 256 beats.
Ø LN 190‐195: According to the sample size calculation, 13 subjects for each group should be sufficient. Why did you recruit 30 subjects per group?
Answer: Thanks for the comment. In a study published in 2013, Raimundo et al studied 38 stroke patients recorded using a heart rate (HR) monitor and the data were used to assess cardiac autonomic modulation through HRV analysis. The control group was later collected, this increased the power of the test.
- Raimundo, R.D., de Abreu, L.C., Adami, F. et al. Heart Rate Variability in Stroke Patients Submitted to an Acute Bout of Aerobic Exercise. Transl. Stroke Res. 4, 488–499 (2013)
4.Results:
Ø In table 3 and 4, the meaning of symbols should be added, such as “#”, “§”, etc.
Answer: Thanks for the comment. All abbreviations, signs and asterisks have been revised and inserted into the caption
Ø Please add the results of IPAQ from the subjects to Table 1.
Answer: Thanks for the comment The results of IPAQ was added to Table 1
Discussion:
Ø The discussion section is weak and needs to be strengthened up. In addition, what health policy should be implemented and why it is important should be discussed based on the results of this study.
Answer: Thanks for the comment. The discussion was rewritten, strengthening the relationship between the research and public policies.
Ø LN54‐58: How does this paragraph relate to your results? Please discuss.
Answer: Thanks for the comment. The paragraph has been rewritten. All changes are in red.
Ø LN35‐41: The coherence of this paragraph is weak.
Answer: Thanks for the comment. The paragraph has been rewritten. All changes are in red.
Ø LN46‐53: The paragraph is weak, only the statement of the results.
Answer: Thanks for the comment. The paragraph has been rewritten. All changes are in red.
Ø LN60‐68: Both Ref 11 and Ref 34 investigated the training effect. This study did not involve exercise training. No training effect should be presumed.
Answer: The paragraph has been supplemented.
Ø Were all of the stroke subjects chronic patients? Will the results of this study apply to subacute or acute stroke patients? Please discuss.
Answer: This comment was placed on study limitations. Our stroke patients were chronic, so our results cannot be extrapolated to acute and subacute patients.
Ø This study is to compare autonomic modulation between elderly stroke patients and healthy elderly individuals at rest, during single bout of exercise, and post exercise. What do the results contribute to public health policy? What are the results relevant to clinical application? Please discuss.
Answer: The last paragraph was rewritten.
Rehabilitation services provide the management of neuromuscular and musculoskeletal disorders that alter functional status. Major weaknesses are described in the implementation of public policies for the rehabilitation care of stroke patients. Public health services must be prepared and integrated into public policies on disability. In conclusion, the present study demonstrated that elderly patients with stroke have reduced variability at rest, sympathetic predominance during exercise and do not return to baseline values over the 30 minutes of recovery, with similar responses found in the group of healthy elderly people. According to WHO, rehabilitation are measures that help people with disabilities to maintain optimal functionality and should be a strategy for equalizing opportunities and social integration, monitoring the autonomic nervous system can be a cheap and non-invasive tool to help prescribe exercises for this population. Longitudinal studies with stroke patients can elucidate the role of HRV in the prescription and safety of aerobic exercise and, perhaps, provide subsidies to promote an improvement in the quality of rehabilitation provision.
.Ø What are the limitations of this study?
Answer: Thanks for the comment. A paragraph of study limitations was inserted

Reviewer 2 Report
This study was to investigate the differences on autonomic modulation between
elderly adults with chronic stroke and healthy controls at rest, during exercise and
post exercise. There are my comments.
1. The title does not appear to reflect the main purpose of this study.
2. Introduction:
It has been known that aerobic exercise training is beneficial to individuals after
stroke, including elderly stroke patients. The justification of this study should be
strengthened up. For example, why is it important to explore the autonomic
modulation during exercise and post exercise, especially from the perspective of
public health policy?
LN73: It is inappropriate to cite Ref 3, 4, and 12. For example, Ref 12 did not
involve exercise training.
LN 91‐94: The function of autonomic nervous system is to help our body to
response to external stimuli, such as exercise. Regarding the hypothesis, what do
you mean by “an acute bout of aerobic exercise would be able to promote
changes in cardiac autonomic modulation of elderly stroke patients”? Please
clarify.
3. Method:
LN104‐110: Please specifically list your inclusion and exclusion criteria for
stroke participants and healthy participants.
LN126‐127: Please provide the cutoff score for classifying the level of
physical activity. In addition, is IPAQ valid for stroke population?
LN 127‐130: Brief introductions on those assessment tools should be added.
LN137‐140: what kind of equation did you use to calculate age‐predicted
maximal heart rate?
LN174‐176: How many data were excluded? Please report this information
in the result section.
LN 180‐188: This paragraph is confusing. Especially, the statement, “The
secondary outcome was to compare cardiac autonomic modulation of
elderly people with stroke with general population.”, should not this be the
main purpose of this study?
LN 190‐195: According to the sample size calculation, 13 subjects for each
group should be sufficient. Why did you recruit 30 subjects per group?
4. Results:
In table 3 and 4, the meaning of symbols should be added, such as “#”, “§”,
etc.
Please add the results of IPAQ from the subjects to Table 1.
5. Discussion:
The discussion section is weak and needs to be strengthened up. In addition,
what health policy should be implemented and why it is important should
be discussed based on the results of this study.
LN54‐58: How does this paragraph relate to your results? Please discuss.
LN35‐41: The coherence of this paragraph is weak.
LN46‐53: The paragraph is weak, only the statement of the results.
LN60‐68: Both Ref 11 and Ref 34 investigated the training effect. This study
did not involve exercise training. No training effect should be presumed.
Were all of the stroke subjects chronic patients? Will the results of this study
apply to subacute or acute stroke patients? Please discuss.
This study is to compare autonomic modulation between elderly stroke
patients and healthy elderly individuals at rest, during single bout of
exercise, and post exercise. What do the results contribute to public health
policy? What are the results relevant to clinical application? Please discuss.
What are the limitations of this study?
Author Response
To: Aerobic exercise in elderly people with stroke as a public health policy
Manuscript ID: ijerph-1385475
Type of manuscript: Article
Authors: Rodrigo Daminello Raimundo, Juliana Zangirolami-Raimundo, Claudio Leone, Tatiana Dias de Carvalho, Talita Dias da Silva, Italla Maria Pinheiro Bezerra, Alvaro Dantas de Almeida-Junior, Vitor Engracia Valenti, Luiz Carlos de Abreu
Dear Dr.
We very much appreciate you for your highly constructive reviews to our submission. We have revised the material to eliminate the issues raised. The added or modified words, phrases, and sentences are in red. We hope the present version can be accepted.
Review 1
Thank you for submitting the manuscript.
The aim of this study was to evaluate the changes in heart rate variability of elderly healthy subjects and stroke patients. The results show reduced HRV in elderly stroke group. This study raises some other questions and comments for me. In particular, I think the inadequate description of stroke patients is especially relevant. It can be generally said that about 1/3 of the stroke patients have no residuals at all. The lack of interpretation and comparability with the control group limited the assessment of HRV.
Answer:
Thank you for the attention on our manuscript. As suggested, we have re-written the manuscript in order to improve it. We really appreciate the kind suggestions and recommendations of the reviewers, which significantly helped us to improve the manuscript. We hope the present version can be accepted for publication on this very important journal.
General comments
- The reference list does not correspond to one of the two recommended reference styles. The author guidelines should be read and the reference list should be adjusted. Use [ ] for citation in the text.
Answer: Reference list and style were adjusted (Mendeley reference manager)
- Under all tables, explain abbreviations, signs and the asterisks when listing statistics. For example, what does § mean in table 4? Please check all tables regarding other characters.
Answer: Thank you for your comment. All abbreviations, signs and asterisks have been revised and inserted into the caption
- All variables and R3, main effect etc. should be marked in bold. Please check all tables.
Answer: Thank you for your comment. All variables and R3, main effect were marked in bold
Introduction
- The reader is not introduced to the method of HRV at all. It remains unclear what HRV means. How does a normal stress reaction proceed in healthy people; in this example under physical stress? What is known about this in stroke patients? A reader who is not familiar with HRV cannot do anything at all with the meaning of the data. Please add to that. Without this information, the meaning of your work cannot be explained to every reader.
Answer: Thank you for taking the time to send us your feedback. The introduction was rewritten according to reviewers' comments. All changes are in red.
- What HRV guidelines were used? Which reference values for the HRV parameters were used? Which confounders can influence HRV?
- I recommend common guidelines such as:
- Task Force of the European Society of Cardiology and the North American Society of Pacing and Electrophysiology (1996): Heart rate variability. Standards of measurement, physiological interpretation, and clinical use. 0041. In: European heart journal 17 (3), S. 354–38
- Sammito, Stefan; Thielmann, Beatrice; Seibt, Reingard; Klussmann, André; Weippert, Matthias; Böckelmann, Irina (2015): Guideline for the application of heart rate and heart rate variability in occupational medicine and occupational science. 0651. In: ASU International. DOI: 10.17147/ASUI.2015-06-09-03.
Answer:
Thank you for your comment. References cited by reviewers were used to define HRV guidelines. Reference values cited by Sammito et al, as well as confounding factors, were incorporated into the discussion.
- Your national guideline, if applicable.
Answer: Yes, these are the references used in Brazil and that were used in the research
Methods
- The citation for the reference to the STROBE guidelines is missing and should be added.
Answer: Thanks for remembering that. The reference was added
von Elm E, Altman DG, Egger M, Pocock SJ, Gøtzsche PC, Vandenbroucke JP; STROBE Initiative. The Strengthening the Reporting of Observational Studies in Epidemiology (STROBE) statement: guidelines for reporting observational studies. Lancet. 2007 Oct 20;370(9596):1453-7. doi: 10.1016/S0140-6736(07)61602-X. PMID: 18064739.
- STROBE guidelines also include the timeline of investigations. No information is provided on this. Please add the time period of the examinations.
Answer: Thank you for your comment. Various information has been added in the method section.
- Study population: I can't believe that the stroke patients didn't take any medications that affect HRV. What about hypertension and antihypertensive drugs? Cardiac arrhythmias and hypertension are common causes in stroke patients. I request a comment on hypertension and drugs in both groups because hypertension decreases HRV. Was it comparable? If it was not collected, it should be included as a limitation of the study.
Answer: Thank you for your comment. The collection of this study took a long time due to the healthy elderly group, it was very difficult to get a number of elderly people who did not use drugs for arrhythmias, hypertension or any other metabolic disease but after 4 years we got these data. As for the stroke group, we did not include patients who used beta-blockers. Other medications for arrhythmia and hypertension were not asked in the initial anmesis, we inserted this in the limitations.The information below was included in the discussion:
“The inclusion of stroke patients without beta-blockers limits the external validation of this work due to the large use of this drug by this population, in addition, it was not collected in the initial anamnesis if the stroke patients used other medications for hypertension”
- Shouldn’t the duration of stroke and elevation be more detailed? What is the functional status of stroke patients? In general, it can be said that 1/3 of the stroke patients have no residuals anymore.
Answer: Thank you for your comment. More details about the assessment instruments have been added. The Fugl-Meyer assessment is widely applied in international experiments aimed at individuals with hemiparesis. Tests provide conditions for scoring functional activities. It is divided into five domains: motor function, sensitivity, balance, range of motion and pain. The domain of motor function includes measurement of movement, coordination and reflex activity of the shoulder, elbow, wrist, hand, hip and ankle, totaling 100 points, with 66 referring to the upper extremity and referring to the lower extremity. Depending on the total score, the patient can be classified as having severe, moderate or mild impairment.
- How should the data from Table 1 (e.g., MMSE) be interpreted? Are the data good or poor? I can well imagine that a greater loss of function also means more physical stress and thus can influence the HRV parameter.
Answer: Thank you for your comment. The MMSE scale was detailed in the method. The MMSE is a test used to assess cognitive function because it is fast. It is divided into domains and divided into two steps. The first seeks to assess orientation, memory and attention and the second analyzes specific skills such as naming and comprehension (30 points in total). The higher a patient's score, the better their cognitive performance. A patient who scores more than 25 points is considered normal. Mild cognitive loss is suspected when the score is between 21 and 24 points, moderate, between 10 and 20, and severe less than or equal to 9. [22]
- Was the training load of both groups the same?
Answer: Thank you for your comment. The information was added. After the initial evaluation the heart monitor strap was placed on each subject’s thorax over the distal third of the sternum. Measurements were subsequently obtained at rest before starting the exercise on the respective bench press for 10 minutes (min), during incremental aerobic exercise for 30 min and more 30 min bench press post exercise (recovery period). The stipulated training load for both groups was 60-70% of the maximum HR (HRmax). Based on Tanaka et al. the HRmax calculated was HRmax = 207 – 0.7 × age.
Tanaka H, Monahan KD, Seals DR: Age-predicted maximal heart rate revisited. J Am Coll Cardiol 2001; 37: 153-156.
- Ethics statement: It should be added that the Declaration of Helsinki with its ethical guidelines of medical research on humans was observed.
Answer: Thanks for remembering that. All subjects gave their written informed consent for this study, which was approved by Research Ethics Committee (CAEE: 43334714.6.0000.5429, permit number 1.017.631). The Declaration of Helsinki with its ethical guidelines of medical research on humans was observed.
- Initial assessment: An interpretation should be presented to evaluate the values of Table 1. Why are these missing for the control group? Please add to establish comparability. If these data are missing, it should be listed as a limitation of the study. In principle, it is conceivable that the stroke patients no longer have residuals and thus have good function similar to the control group.
Answer: Thanks for the comment, but table 1 has no data missing. All possible comparisons have been made. Data as Type Injury (%, ischemic); Hemisphere of injury (%, left) and Time injury (years) cannot be compared as the control group has no neurological damage. Also, functionality scales such as Fugl-Meyer and Orpington severity are not used on uninjured people. Comments from the Mini-Mental State Examination were inserted in the study limitation.
- Assessment of HRV: The comparability of HRV data from different recording lengths is methodologically objectionable and not meaningful. It should be corrected. The analysis method in frequency domain parameters should be supplemented (e.g. Fast Fourier Transform, Autoregressive Model, Trigonometric Regressive Spectral Analysis). FFT analysis requires stationarity and is not present in the exercises.
Answer: Thanks for the comments. The evaluation of HRV performed in this research following the norms of the Task Force of the European Society of Cardiology and the North American Society of Pacing and Electrophysiology (Circulation 1996). The authors of this manuscript (Rodrigo Daminello Raimundo, Vitor Engracia Valenti and Luiz Carlos de Abreu) total more than 200 publications on the subject.
In order to provide further details regarding HRV analysis, we added the following sentence:
Method, 2.3.2, 5th paragraph:
“We calculated the spectral analysis using the Fast Fourier Transform algorithm. We selected stable series during exercise.”
https://pubmed.ncbi.nlm.nih.gov/?term=Valenti+VE&sort=pubdate
https://pubmed.ncbi.nlm.nih.gov/?term=Abreu+LC&sort=pubdate
https://pubmed.ncbi.nlm.nih.gov/?term=Raimundo+RD&sort=pubdate
https://pubmed.ncbi.nlm.nih.gov/?term=Daminello-Raimundo+R&sort=pubdate
Discussion
- The discussion should be revised. E.g. the 1st paragraph can be moved to the introduction (see comment 3).
Answer: Thanks for the comment. The first paragraph was sent for the introduction and complemented by another previous paragraph as "comment 3"
- Lines 46-48: Which reference values are referred to here?
Answer: Thanks for the comment. Reference values in HRV are still questionable, so the option of this research was to use a control group as similar as possible, even so, the references below were inserted and used in the discussion. Although reference values in HRV are still discussed, some studies have tried to make data for comparison possible. Our results showed, when compared with normality values from the researches by Sammito & Böckelmann and Dantas et al higher values of SDNN and lower values of pNN50, rMMSD, LF, HF and LF/HF when compared to same age group.
- Sammito S, Böckelmann I. New reference values of heart rate variability during ordinary daily activity. Heart Rhythm. 2017 Feb;14(2):304-307. doi: 10.1016/j.hrthm.2016.12.016. Epub 2016 Dec 14. PMID: 27986556.
- Dantas EM, Kemp AH, Andreão RV, da Silva VJD, Brunoni AR, Hoshi RA, Bensenor IM, Lotufo PA, Ribeiro ALP, Mill JG. Reference values for short-term resting-state heart rate variability in healthy adults: Results from the Brazilian Longitudinal Study of Adult Health-ELSA-Brasil study. Psychophysiology. 2018 Jun;55(6):e13052. doi: 10.1111/psyp.13052. Epub 2018 Jan 2. PMID: 29292837.
- Sammito S, Böckelmann I. Reference values for time- and frequency-domain heart rate variability measures. Heart Rhythm. 2016 Jun;13(6):1309-16. doi: 10.1016/j.hrthm.2016.02.006. Epub 2016 Feb 12. PMID: 26883166.
- What is the new knowledge in addition to the systematic review?:
- Yperzeele L, van Hooff R-J, Nagels G et al. (2015) Heart rate variability and baroreceptor sensitivity in acute stroke: a systematic review. Int J Stroke 10: 796–800. https://doi.org/10.1111/ijs.12573
Answer: Thanks for the comment. We understand that the review carried out by Yperzeele et al kept doubts due to “varied methodology, small samples, survival selection and exclusion of patients with frequent comorbidities in stroke”. Our study tries to compare stroke patients with a group of elderly people without cardiovascular and metabolic alterations (which is rare). Data collection for this group took a long time due to this fact. We believe that comparing a group with these characteristics is new.
Information from the work of Yperzeele et al was added to the discussion.
- The discussion should end with a conclusion: what is the clinical relevance of the study? Which measures are recommended for the rehabilitation of patients or for further research?
Answer: Thanks for the comment. The last paragraph was rewritten. Rehabilitation services provide the management of neuromuscular and musculoskeletal disorders that alter functional status. Major weaknesses are described in the implementation of public policies for the rehabilitation care of stroke patients. Public health services must be prepared and integrated into public policies on disability. In conclusion, the present study demonstrated that elderly patients with stroke have reduced variability at rest, sympathetic predominance during exercise and do not return to baseline values over the 30 minutes of recovery, with similar responses found in the group of healthy elderly people. According to WHO, rehabilitation are measures that help people with disabilities to maintain optimal functionality and should be a strategy for equalizing opportunities and social integration, monitoring the autonomic nervous system can be a cheap and non-invasive tool to help prescribe exercises for this population. Longitudinal studies with stroke patients can elucidate the role of HRV in the prescription and safety of aerobic exercise and, perhaps, provide subsidies to promote an improvement in the quality of rehabilitation provision.
- Study limitations should be listed and discussed. Currently, some limitations are found, e.g., different duration of HRV recording between phases, lack of specification of the analysis method for the frequency domain, insufficient description of the functional status of stroke patients, period of stroke and study period, hypertension history of stroke patients, and intake of antihypertensive medication.
Answer: Thanks for the comment. A paragraph on study limitation was inserted.
Our study has some limitations. The inclusion of patients with stroke without beta-blockers limits the external validation of this research due to the large use of this drug by this population, in addition, it was not collected in the initial anamnesis if the patients with stroke used other medications for hypertension. Despite not being the objective of the study, another limitation was the non-application of the MMSE in the control group, thus, there is no possibility of comparing the cognitive status of the elderly.
Review 2
Dear Dr.
We very much appreciate you for your highly constructive reviews to our submission. We have revised the material to eliminate the issues raised. The added or modified words, phrases, and sentences are in red. We hope the present version can be accepted.
Comments and Suggestions for Authors
This study was to investigate the differences on autonomic modulation between elderly adults with chronic stroke and healthy controls at rest, during exercise and post exercise. There are my comments.
The title does not appear to reflect the main purpose of this study.
Answer: Thanks for the comment. The title was changed.
The use of cardiac autonomic responses to aerobic exercise in elderly stroke patients: functional rehabilitation as a public health policy
2.Introduction:
- It has been known that aerobic exercise training is beneficial to individuals after stroke, including elderly stroke patients. The justification of this study should be strengthened up. For example, why is it important to explore the autonomic modulation during exercise and post exercise, especially from the perspective of public health policy?
Answer: Thank you for the attention on our manuscript. As suggested, we have re-written the manuscript in order to improve it. We really appreciate the kind suggestions and recommendations of the reviewers, which significantly helped us to improve the manuscript.
The introduction and discussion of the manuscript were rewritten, strengthening the importance of exploring autonomous modulation during exercise and post-exercise from the perspective of public health policy. All changes are in red.
Ø LN73: It is inappropriate to cite Ref 3, 4, and 12. For example, Ref 12 did not involve exercise training.
Answer: Thanks for the comment. References have been changed
Ø LN 91‐94: The function of autonomic nervous system is to help our body to response to external stimuli, such as exercise. Regarding the hypothesis, what do you mean by “an acute bout of aerobic exercise would be able to promote changes in cardiac autonomic modulation of elderly stroke patients”? Please clarify.
Answer: Thanks for the comment, the sentence did not reflect the study hypothesis. The paragraph has been rewritten. Thus, the need for effective stroke rehabilitation is likely to remain an essential part of the stroke care continuum for the foreseeable future and in improving health policies for functional rehabilitation. However, there are few studies comparing an elderly stroke population to healthy elderly control subjects. Our hypothesis is that an acute session of aerobic exercise would be able to promote different changes in the cardiac autonomic modulation of healthy elderly people with elderly people with stroke. Therefore, the aim of this study is to compare cardiac autonomic modulation in elderly stroke patients before, during and after an acute session of aerobic exercise.
Method:
Ø LN104‐110: Please specifically list your inclusion and exclusion criteria for stroke participants and healthy participants.
Answer: Thank you for your comment. Various information has been added in the method section.
Ø LN126‐127: Please provide the cutoff score for classifying the level of physical activity. In addition, is IPAQ valid for stroke population?
Answer: Thank you for your comment. More details about the assessment instruments have been added. Information about the IPAQ has been inserted into the method. The IPAQ is used for the stroke population as well. The IPAQ is a validated instrument that allows estimating the weekly time spent on physical activities. Possible categories are: "Sedentary" (does not perform any physical activity for at least 10 continuous minutes during the week); "Insufficiently Active" (individuals who practice physical activities for at least 10 continuous minutes per week, but not enough to be classified as active) and "Active" or "Very Active" (comply with weekly exercise time recommendations. The IPAQ is validated for stroke patients
Ruescas-Nicolau MA, Sánchez-Sánchez ML, Cortés-Amador S, Pérez-Alenda S, Arnal-Gómez A, Climent-Toledo A, Carrasco JJ. Validity of the International Physical Activity Questionnaire Long Form for Assessing Physical Activity and Sedentary Behavior in Subjects with Chronic Stroke. Int J Environ Res Public Health. 2021 Apr 29;18(9):4729. doi: 10.3390/ijerph18094729. PMID: 33946690; PMCID: PMC8125179.
Ø LN 127‐130: Brief introductions on those assessment tools should be added.
Answer: Thank you for your comment. The method has been rewritten with the necessary information.
Ø LN137‐140: what kind of equation did you use to calculate age‐predicted maximal heart rate?
Answer: Thank you for your comment. The method has been rewritten with the necessary information. After the initial evaluation the heart monitor strap was placed on each subject’s thorax over the distal third of the sternum. Measurements were subsequently obtained at rest before starting the exercise on the respective bench press for 10 minutes (min), during incremental aerobic exercise for 30 min and more 30 min bench press post exercise (recovery period). The stipulated training load for both groups was 60-70% of the maximum HR (HRmax). Based on Tanaka et al. the HRmax calculated was HRmax = 207 – 0.7 × age.
References:
- Sammito, Stefan; Thielmann, Beatrice; Seibt, Reingard; Klussmann, André; Weippert, Matthias; Böckelmann, Irina (2015): Guideline for the application of heart rate and heart rate variability in occupational medicine and occupational science. 0651. In: ASU International. DOI: 10.17147/ASUI.2015-06-09-03.
- Camarda SR, Tebexreni AS, Páfaro CN, Sasai FB, Tambeiro VL, Juliano Y et al. Comparison of maximal heart rate using the prediction equations proposed by Karvonen and Tanaka. Arq Bras Cardiol. 2008;91:311–4.
- Robergs RA, Landwehr R. The surprising history of the Hrmax =“220-age” equation. J Exerc Physiol. 2002;5(2):1–10.
- Tanaka H, Monahan KD, Seals DR. Age—predicted maximal heart revisited. J Am Coll Cardiol. 2001;37:153–6.
Ø LN174‐176: How many data were excluded? Please report this information in the result section.
Answer: Thank you for your comment. The results was rewriten. Patients from a neurology outpatient clinic were recruited into the stroke group. Of a total of 45 stroke patients, 30 were selected (five did not want to participate in the study - difficulty walking to the exercise location and 10 did not meet the eligibility criteria - eight used beta-blockers and two used antiarrhythmic drugs). For the group of general population were re-cruited from the community who volunteered for an exercise program. These elderly people were called for initial assessment and if they ac-cepted to participate in the study they were included. 30 elderly people were initially recruited, but six had diabetes mellitus and / or systemic arterial hypertension, after this pre-selection, 10 more patients were re-cruited and 4 did not want to participate in the study.
Ø LN 180‐188: This paragraph is confusing. Especially, the statement, “The secondary outcome was to compare cardiac autonomic modulation of elderly people with stroke with general population.”, should not this be the main purpose of this study?
Answer: Thanks for the comment. The outcomes are reversed.
The primary outcome was to compare cardiac autonomic modulation of elderly people with stroke with the general population. The secondary outcome was measured in: Rest- 10 min of rest in supine, Exercise- the 20 minutes of peak exercise (removing the first and the last 5 min of 30 min exercise) and Recovery- 30 min in supine post exercise. The post exercise period was divided into (R1) first 5 min of recovery; (R2) half of the recovery time, the period between the twelfth- (12th) and seventeenth- (17th) minute recovery; (R3) final 5 min of recovery. At all times the amount of RR exceeded 256 beats.
Ø LN 190‐195: According to the sample size calculation, 13 subjects for each group should be sufficient. Why did you recruit 30 subjects per group?
Answer: Thanks for the comment. In a study published in 2013, Raimundo et al studied 38 stroke patients recorded using a heart rate (HR) monitor and the data were used to assess cardiac autonomic modulation through HRV analysis. The control group was later collected, this increased the power of the test.
- Raimundo, R.D., de Abreu, L.C., Adami, F. et al. Heart Rate Variability in Stroke Patients Submitted to an Acute Bout of Aerobic Exercise. Transl. Stroke Res. 4, 488–499 (2013)
4.Results:
Ø In table 3 and 4, the meaning of symbols should be added, such as “#”, “§”, etc.
Answer: Thanks for the comment. All abbreviations, signs and asterisks have been revised and inserted into the caption
Ø Please add the results of IPAQ from the subjects to Table 1.
Answer: Thanks for the comment The results of IPAQ was added to Table 1
Discussion:
Ø The discussion section is weak and needs to be strengthened up. In addition, what health policy should be implemented and why it is important should be discussed based on the results of this study.
Answer: Thanks for the comment. The discussion was rewritten, strengthening the relationship between the research and public policies.
Ø LN54‐58: How does this paragraph relate to your results? Please discuss.
Answer: Thanks for the comment. The paragraph has been rewritten. All changes are in red.
Ø LN35‐41: The coherence of this paragraph is weak.
Answer: Thanks for the comment. The paragraph has been rewritten. All changes are in red.
Ø LN46‐53: The paragraph is weak, only the statement of the results.
Answer: Thanks for the comment. The paragraph has been rewritten. All changes are in red.
Ø LN60‐68: Both Ref 11 and Ref 34 investigated the training effect. This study did not involve exercise training. No training effect should be presumed.
Answer: The paragraph has been supplemented.
Ø Were all of the stroke subjects chronic patients? Will the results of this study apply to subacute or acute stroke patients? Please discuss.
Answer: This comment was placed on study limitations. Our stroke patients were chronic, so our results cannot be extrapolated to acute and subacute patients.
Ø This study is to compare autonomic modulation between elderly stroke patients and healthy elderly individuals at rest, during single bout of exercise, and post exercise. What do the results contribute to public health policy? What are the results relevant to clinical application? Please discuss.
Answer: The last paragraph was rewritten.
Rehabilitation services provide the management of neuromuscular and musculoskeletal disorders that alter functional status. Major weaknesses are described in the implementation of public policies for the rehabilitation care of stroke patients. Public health services must be prepared and integrated into public policies on disability. In conclusion, the present study demonstrated that elderly patients with stroke have reduced variability at rest, sympathetic predominance during exercise and do not return to baseline values over the 30 minutes of recovery, with similar responses found in the group of healthy elderly people. According to WHO, rehabilitation are measures that help people with disabilities to maintain optimal functionality and should be a strategy for equalizing opportunities and social integration, monitoring the autonomic nervous system can be a cheap and non-invasive tool to help prescribe exercises for this population. Longitudinal studies with stroke patients can elucidate the role of HRV in the prescription and safety of aerobic exercise and, perhaps, provide subsidies to promote an improvement in the quality of rehabilitation provision.
Ø What are the limitations of this study?
Answer: Thanks for the comment. A paragraph of study limitations was inserted

Round 2
Reviewer 1 Report
Dear Authors,
thank you very much for the implementation of my comments. The manuscript is thus better understandable. The reference list is not in accordance with the author guidelines. Please adjust it. All the best for your further research.
Author Response
Dear Dr.
We very much appreciate you for your highly constructive reviews to our submission. We have revised the material to eliminate the issues raised. We hope the present version can be accepted.
- Reference list and style were adjusted (Mendeley reference manager)

Reviewer 2 Report
The manuscript has been much improved. There are few suggestions as follows.
LN 163-164: The following sentence is confusing and please confirm it. “Both groups did not included patients who did not have motor conditions to perform the exercise on a treadmill”
LN 76 in discussion: “60 stroke”, should not it be “30”?
Author Response
Dear Dr.
We very much appreciate you for your highly constructive reviews to our submission. We have revised the material to eliminate the issues raised. We hope the present version can be accepted.
- LN 163-164: The following sentence is confusing and please confirm it. “Both groups did not included patients who did not have motor conditions to perform the exercise on a treadmill”
Answer: Thanks for the comments. The sentence was removed, and the paragraph was adapted.
LN 76 in discussion: “60 stroke”, should not it be “30”?
Answer: Thank you for the attention on our manuscript. The number has been changed.
